# Effects of trust-based decision making in disrupted supply chains

**Rozhin Doroudi**[1]*, **Pedro Sequeira**[2,3], **Stacy Marsella**[3,4], **Ozlem Ergun**[1], **Rana Azghandi**[1], **David Kaeli**[5], **Yifan Sun**[5], **Jacqueline Griffin**[1]

**1** Department of Mechanical and Industrial Engineering, Northeastern University, Boston, MA, United States of America, **2** SRI International, Menlo Park, CA, United States of America, **3** Khoury College of Computer Sciences, Northeastern University, Boston, MA, United States of America, **4** Institute of Neuroscience and Psychology, University of Glasgow, Glasgow, United Kingdom, **5** Department of Electrical and Computer Engineering, Northeastern University, Boston, MA, United States of America

* doroudi.r@northeastern.edu

**Data Availability Statement:** The data underlying the results presented in the study are available from: https://doi.org/10.5281/zenodo.3627633.

**Funding:** This work is supported by the National Science Foundation under grant CMMI 1638302 to

## Abstract

The United States has experienced prolonged severe shortages of vital medications over the past two decades. The causes underlying the severity and prolongation of these shortages are complex, in part due to the complexity of the underlying supply chain networks, which involve supplier-buyer interactions across multiple entities with competitive and cooperative goals. This leads to interesting challenges in maintaining consistent interactions and trust among the entities. Furthermore, disruptions in supply chains influence trust by inducing over-reactive behaviors across the network, thereby impacting the ability to consistently meet the resulting fluctuating demand. To explore these issues, we model a pharmaceutical supply chain with boundedly rational artificial decision makers capable of reasoning about the motivations and behaviors of others. We use multiagent simulations where each agent represents a key decision maker in a pharmaceutical supply chain. The agents possess a Theory-of-Mind capability to reason about the beliefs, and past and future behaviors of other agents, which allows them to assess other agents' trustworthiness. Further, each agent has beliefs about others' perceptions of its own trustworthiness that, in turn, impact its behavior. Our experiments reveal several counter-intuitive results showing how small, local disruptions can have cascading global consequences that persist over time. For example, a buyer, to protect itself from disruptions, may dynamically shift to ordering from suppliers with a higher perceived trustworthiness, while the supplier may prefer buyers with more stable ordering behavior. This asymmetry can put the trust-sensitive buyer at a disadvantage during shortages. Further, we demonstrate how the timing and scale of disruptions interact with a buyer's sensitivity to trustworthiness. This interaction can engender different behaviors and impact the overall supply chain performance, either prolonging and exacerbating even small local disruptions, or mitigating a disruption's effects. Additionally, we discuss the implications of these results for supply chain operations.

JG. (https://www.nsf.gov/awardsearch/showAward?AWD_ID=1638302&HistoricalAwards=false). SRI International provided support in the form of salaries for author P. Sequeira but did not have any additional role in the study design, data collection and analysis, decision to publish, or preparation of the manuscript. The specific roles of the authors are articulated in the 'author contributions' section.

**Competing interests:** SRI International provided support in the form of salaries for author P. Sequeira. This does not alter our adherence to PLOS ONE policies on sharing data and materials.

## Introduction

Over the past two decades there has been an epidemic of drug shortages affecting the United States. According to [1], between 2008 and 2014 the number of lifesaving drugs in shortage increased by 393% and the number of drugs in shortage with no acceptable substitute increased by 125%. According to the American Society of Health-System Pharmacists (ASHP), the rate of new shortages is increasing in recent years and the number of active shortages in the second quarter of 2019 reached a height of 282 products [2]. In a 2018 survey of 719 pharmacy practice managers and pharmacy leaders, 69.2% of them state that in the previous year they experienced more than 50 shortages [3].

Drug shortages can have significant dire consequences such as cancelled surgeries [4] and postponed chemotherapy infusions [5]. In cases where a drug can be substituted with another product, drug shortages result in greater costs incurred by an already burdened healthcare delivery system [6]. Thus, these shortages directly translate into a risk to public health and safety.

Further, the United States is not the only country struggling with drug shortages, as it has become a more global issue. For example, Pauwels et al. [7] sent a survey to Hospital Pharmacy Europe subscribers and 45% stated that they had experienced a shortage of life-saving drugs and 30% of respondents associated these shortages with an increase in healthcare costs. A 2019 article [8] reports that drug shortages in Europe are still persisting.

A variety of interacting stakeholders are involved in production and distribution of pharmaceutical drugs, including manufacturers, distributors, and healthcenters. An example of the interactions between several decision makers in a pharmaceutical supply chain is depicted in Fig 1. Based on the information provided by manufacturers to the University of Utah Drug Information Service, the reasons for 51% of reported drug shortages are unknown [2]. Anecdotal evidence [9] suggests that stakeholders' decision making behaviors have an important role in aggravating drug shortages. In a survey from the Pew Research Center and the International Society for Pharmaceutical Engineering [10], manufacturers claimed that the lack of guaranteed volume contracts (i.e., steady ordering behavior from buyers), is a key barrier for entering a market to resolve existing shortages or putting mechanisms in place to prevent future shortages. On the other hand, the IQVIA Institute for Human Data Science reported that the variability in the amounts of products received by healthcenters, is a "sentinel of problems" with instability in the U.S. pharmaceutical supply chain [11]. To mitigate both supply

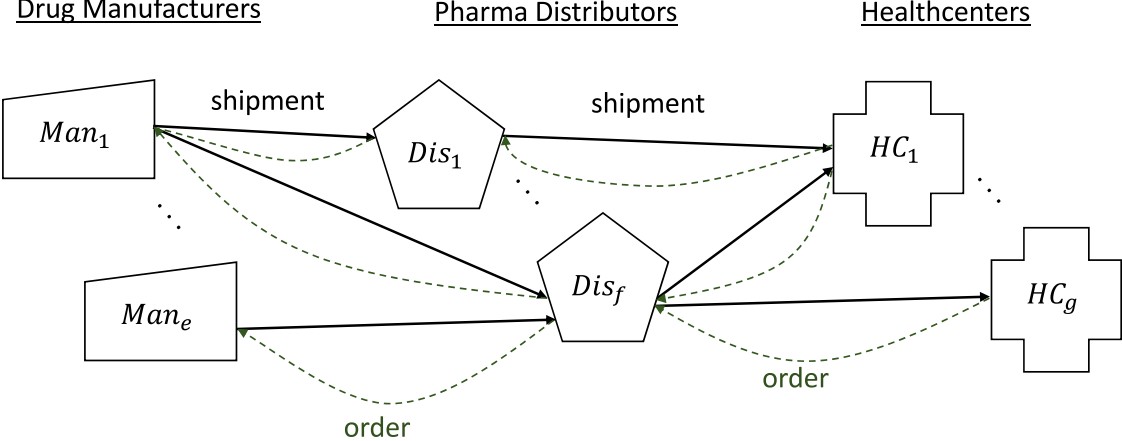

**Fig 1. Interaction between decision makers in a pharmaceutical supply chain.**

and demand uncertainty driving this destabilization, a better understanding of the main drivers of the observed variability and how these drivers contribute to prolonged shortages is needed.

Specifically, there is a need to examine how behaviors of supply chain decision makers and their responsiveness to observations, whether of supply disruptions or others' actions, drive the system's instability. For example, decision makers in a supply chain may change their order amounts based on the perceived trustworthiness of their suppliers, or how reliable their suppliers are in providing the promised amount of product at the promised time. On the other hand, if there is a shortage, suppliers must decide how to allocate the limited inventory based on their beliefs about others' future behaviors as informed by their prior (observed) behavior. This in turn changes the beliefs and decision making of the other supply chain members. We hypothesize that this joint adaptation can destabilize the whole system by propagating and exacerbating the effects of the disruption across the supply chain.

Further, the effects of the disruption may vary based on buyers' sensitivities to changes in on-time delivery rates, or the perceived trustworthiness of their suppliers. Aside from these endogenous behavioral drivers, decision makers also adjust their behaviors in response to exogenous factors such as the scale and timing of disruptions. Thus, we hypothesize that variations in the disruptions' features or in the decision makers' sensitivities to changes in on-time delivery rates can lead to different system outcomes as the effects of the disruptions cascade across the supply chain.

In this paper we focus on three driving factors behind disruptions in pharmaceutical supply chains: (i) joint adaptation of trustworthiness beliefs, (ii) how sensitive beliefs about trustworthiness are to recent actions taken by other agents, and (iii) disruption characteristics. We use multiagent simulations to model supply chains like the one in Fig 1. Each key decision maker is modeled as an autonomous agent that acts based on partial observations of the state of the supply chain, and reasons about other agents' beliefs and behaviors via a Theory-of-Mind (ToM) capability [12]. In particular, each agent has beliefs about the other agents' perceptions of its trustworthiness which, in turn, impact its own behavior.

In our study we observe that a buyer, in order to protect itself from disruptions, may order more from its more trustworthy supplier. However, suppliers, being sophisticated agents themselves, can reason about other agents in the supply chain. In particular, a supplier may determine that it cannot profitably adjust to the ordering behavior of the buyer, which results in it preferring to prioritize fulfilling the requests of another buyer that exhibits a more regular ordering behavior. As a result, this asymmetric response puts the trust-sensitive buyer at a disadvantage in the aftermath of a disruption or when there is a shortage. More importantly, it may create a long-lasting cascading instability in the system. Furthermore, we find that the level of trust sensitivity of an agent can either aggravate or mitigate the effects of a disruption, depending on the underlying features of the disruption. The important implications of these results for supply chain design and operations are discussed in the paper.

## State of the art

While mitigation of supply chain disruptions has been studied extensively, see [13] for a review, the behavior of decision makers in a disrupted supply chain has received less attention. Rong et al. [14] considered the classic *Beer Game* framework in which humans played the roles of different decision makers. They studied the variability of order amounts in a disrupted supply chain and showed that when a disruption occurs, order variability increases as one moves down the supply chain echelons. Sarkar and Kumar [15] also used the *Beer Game* setting to show that sharing information about disruptions reduces order variability and supply chain

costs. Whereas these papers experiment with human decision makers in a supply chain with only one agent in each echelon, we employ a simulation model with artificial decision makers which are capable of mimicking human decision makers' reasoning about the system evolution and others' behaviors. Additionally, we analyze human decision making within a network with multiple agents in each echelon. This allows us to study a range of behaviors in different roles and with varying numbers of available decisions and reasoning capabilities. This more complex setting is essential to explain real-world supply chains as it allows for modeling decisions regarding splitting orders among suppliers or allocating limited supplies to fulfill different customers' orders. Such decisions highly influence the system performance when there is an ongoing shortage.

There exists an extensive body of literature studying how suppliers allocate their inventory to multiple buyers (e.g. [16–18]) and how buyers split their orders among multiple suppliers (e.g. [19]). Our main focus in this paper is not comparing different ordering or allocation schemes, but, instead, we aim to understand the interplay of these decisions by suppliers and buyers while they act in a network with other decision makers.

Supply chains consist of multiple decision makers which interact in a multi-tiered network structure. Supply chain decision makers possess all the main characteristics that are attributed to agents in multiagent frameworks [20]. Namely, they make their decisions with no or little direct intervention from other decision makers (autonomy). They interact with other decision makers when they are placing orders, allocating products, etc. (social ability). They perceive changes in the supply chain environment and respond in a timely fashion, e.g. they observe an increase in demand and ramp up their production level or order amount (reactivity). Finally, they have goal-directed behaviors that are not just responses to changes in environment, e.g. they launch a new product or introduce new machines to their production line (pro-activeness). Multiagent frameworks have been shown to be appropriate for the study different interactions between supply chain decision makers [21–23] and therefore we adopt a multiagent framework in this paper.

Kimbrough et al. [24] used a multiagent system framework for a *Beer Game* setting, where they examined whether artificial agents could outperform the decision making of humans. They compared results from their model with the best available analytical solutions and showed that the model identified optimal or near optimal solutions. Ghadimi et al. [25] used a multiagent system to study a sustainable supplier selection and order allocation problem. They observed that considering sustainability factors can lead to a longer term relationship between the supplier and buyer. Fu and Fu [26] considered a framework which incorporates both a multiagent system and context-aware computing to optimize cost management in supplier-buyer interactions. They showed that the framework can enhance coordination between the supplier and buyer and strengthen their adaptability. While we also examine a supplier-buyer dyad, our agents can reason not only about future states of the system but also about future behaviors of other agents via ToM reasoning, thereby allowing them to adapt their behavior.

Further, we aim to study joint adaptation of decision makers behaviors when the supply chain is facing a disruption. In that regard, Giannakis and Louis [27] developed a multiagent framework to manage and mitigate supply chain disruptions. While they focused on designing a generic framework that facilitates communication between supply chain agents, in this paper we take a closer look at behavioral dynamics of a supplier-buyer dyad in a disrupted supply chain. We also incorporate disruption profiles as a complicating factor. We argue that these profiles interact with endogenous behavioral dynamics and should be taken into account when designing mechanisms for mitigating effects of disruptions.

Regarding the emergence and impact of trust in supply chain decision making, a few papers have used multiagent simulation models to study these phenomena. These papers

use a definition of trust which is similar to ours, i.e., the ability of a supplier to provide the promised amount of product within the promised time interval. Simulating the *Beer Game* setting, [28] and [29] studied trust dynamics in a multi-echelon supply chain with more than one agent in each echelon. Kim [28] observed that the existence of symmetrical trust results in the emergence of collaboration and that such trust-based relationships reduce inventory variability. Jalbut and Sichman [29] studied the effect of trust, suggestions and lying. They observed that when there is honesty and communication, trust-based relationships thrive. Sen et al. [30] designed adaptive budgeted multi-armed bandit algorithms to discriminate between suppliers with different trustworthiness, by simultaneously managing the budget for exploration and estimation of the reliability of suppliers. A common characteristic of these work is that they only study the trustor's behavior, i.e., how agents trust others. In contrast, due to the joint adaptation of decision makers' behaviors in a supply chain setting, we examine the interplay of both trustor and trustee agents. In addition, our supply chain agents are endowed with a ToM capability to be able to reason about behaviors of other agents in the supply chain. Another important distinction of our study is the inclusion of disruptions, adding a layer of complexity and realism to the study of trust dynamics in supply chains.

Previously, we developed a framework for studying resiliency of critical supply chains considering human behavior [31]. This framework can be used to instantiate and test different disruption scenarios while considering human behavior. We provided some preliminary results to showcase how considering human behavior adds a layer of complexity and realism to studying supply chain disruptions. Building upon that framework, this paper studies a variety of scenarios to understand driving factors of pharmaceutical drug shortages.

## Materials and methods

### Supply chain model

In our simulations, we model a pharmaceutical supply chain consisting of three echelons: *manufacturers*, *distributors* and *healthcenters*. At each time-step, healthcenters serve each unit of patient demand with one unit of product. Healthcenters procure products from distributors. Similarly, distributors procure products by ordering from manufacturers. Based on the orders from the distributors, manufacturers choose an amount to produce in each time-step. To account for product manufacturing, processing, and shipment, a lead time of a predefined number of weeks occurs between placing orders and their fulfillment. For every unit of undistributed inventory, each agent will incur a certain per-unit inventory cost in each time-step. Similarly, if agents are not able to satisfy their customers' orders (from distributor/healthcenter agents) or demand (from patients), a backlog cost will be incurred for each unit of unmet demand in each time-step. Any unmet demand is added on to the backlog to be satisfied in the future, i.e., it will accumulate.

### Decision making

Agents in different echelons have different types of decisions to make, as listed in Table 1. For each type of decision, agents have different decision calculations available to them. Each decision calculation prescribes a procedure that agents follow to make a certain decision, i.e. how much to allocate/order and to/from whom. Manufacturers only have one decision to make, which is to decide how much to produce based on a base-stock decision calculation. The base-stock decision calculation corresponds to an *up-to level* production decision that determines the amount of inventory that agents keep to account for uncertainties in demand and the expected demand during the lead time [32]. In contrast, distributors and healthcenters have two types of

**Table 1. Agent decision types and available decision calculations for each decision type.**

| Agents | Decision Type | Available Decision Calculations |
|---|---|---|
| *Manufacturers* | Production | Base-stock |
| *Distributors* | Order Amount | Base-stock |
| | Inventory Allocation | Proportional |
| | | Preferential |
| *Healthcenters* | Order Amount | Base-stock |
| | Splitting Order | Equally |
| | | Based on trustworthiness |

decisions to make. Namely, for distributors, they need to first order from manufacturers where the total order amount is determined by a base-stock decision calculation followed by an allocation of products to healthcenters. When distributors do not have enough inventory, they need to choose between two types of allocation strategies, either *proportionally*, based on the relative demand from the healthcenters, or *preferentially*, which first allocates inventory to meet the demand of the preferred healthcenter and then distributes the remainder to the next preferred healthcenter. As for healthcenters, they also decide their order amount based on a base-stock decision calculation but have two options for splitting the order amount among the distributors, either *equally*, i.e., ordering the same amount from each distributor, or proportionally, *based on the trustworthiness* attributed to each distributor.

## PsychSim and Theory-of-Mind reasoning

A key attribute of human interactions is that people have the capacity to maintain beliefs about the beliefs, motivations and behaviors of other people involved in the interaction and use those beliefs to inform their own behavior. This capacity to maintain beliefs about others is often referred to as a Theory-of-Mind (ToM) [12]. Decision makers' interactions in a supply chain setting are no exception. While making decisions, each decision maker will account for how others in the supply chain may react to its decision and how their beliefs and behaviors will change as a result. In order to account for this, we use the PsychSim simulation tool [33] that allows for modeling interactions among individuals or groups. PsychSim agents are boundedly-rational decision makers that try to achieve their goals by choosing actions that take into account how the whole environment will evolve. This includes not only their observations over the state of the system but also how other agents may react to their actions. This is achieved by having each agent update its internal models of other agents in the world, i.e., they have a ToM capability.

In our simulations, agents' actions involve realistic production, ordering, and allocation procedures that are computationally-efficient piecewise-linear abstractions of standard predefined decision calculations used by real-world decision makers found in the supply chain literature (e.g. see [32]), as stated in Table 1. An agent's model of other agents consists of its belief about their state, reward function and available actions. While making decisions, PsychSim agents perform ToM reasoning by planning for a certain number of periods into the future. By doing this, they can reason about how other agents will react in any of the hypothetical future states resulting from the possible combinations of all agents' actions. Specifically, agents try to maximize their reward assuming that others also try to maximize their own reward, informed by their current model of other agents. In addition, a discount factor models the fact that a future reward is worth less than an immediate reward. A more detailed explanation of evolution of the system is described in the next section.

## Mathematical model

Supply chain agents act in an environment that dynamically unfolds based on uncertainties concerning supply and demand as well as the beliefs and behaviors of other agents. Therefore, we model our multiagent supply chain as a *Partially Observable Markov Game* (POMG) [34] of interdependent decision makers. In our model, a time-step corresponds to a period of interaction in the supply chain, e.g., a day or a week. A *K*-player POMG evolves as follows:

- At each discrete time-step $t$, the game is in a state $s(t)$ from a finite set $S$ of possible states. The state corresponds to a combination of states of all agents in the supply chain, including order and demand amounts, inventories, backlogs, production levels, etc.

- Each agent $k$ has access to a partial view of $s(t)$, referred to as the observation, denoted as $o_k(t)$, of agent $k$ at time-step $t$. Based on its observations, each agent maintains a probability, referred to as a belief, denoted by $b_k(s)$, of being at state $s$. In this paper, we assume that agents have a perfect model of other agents' states.

- At time-step $t$, each agent $k$ selects, simultaneously and without explicit communication, an action $a_k(t)$ from a set $A_k$ of possible actions. Manufacturers only have a production decision and only a base-stock decision calculation available for them, resulting in one possible *action*. Distributors have two types of decisions to make, an order amount decision and an inventory allocation decision. Therefore they have combined actions which are comprised of a base-stock decision calculation and either allocating the inventory proportionally among healthcenters or preferring one healthcenter to allocate available inventory to that healthcenter first, before allocating inventory to the others. Healthcenters also have two types of decisions to make, the order amount and the ordering split. Thus, their available actions are combined actions comprised of a base-stock decision calculation and either splitting the order equally or splitting the order based on the trustworthiness attributed to distributors. We write $\mathbf{a}(t) = \{a_1(t), \ldots, a_K(t)\}$ to denote the joint action of all agents at $t$.

- Each agent $k$ is awarded a reward $r_k(s(t))$ that depends only on $s(t)$ for some bounded real-valued reward function $r_k : S \to \mathbb{R}$. The reward function encodes the agent's goals, i.e., it returns a scalar denoting how important some state is. Agents, depending on the echelon they belong to, have different reward functions. Manufacturers and distributors try to minimize the sum of their inventory and backlog costs while maximizing the amount they ship to downstream agents. On the other hand, healthcenters try to minimize the sum of their inventory and backlog costs while maximizing the number of patients that they treat.

- The world's dynamics are modeled by a stationary deterministic transition function $T: S \times \mathbf{A} \to S$ where $\mathbf{A} = \langle A_1, \ldots, A_K \rangle$ is the set of all possible joint-actions of the agents in the network. The next state of the supply chain, $s(t + 1)$ depends only on $\mathbf{a}(t)$ and $s(t)$ as dictated by $T$.

In the reported experiments, the world is composed of $K = 6$ agents, each modeled as an individual decision maker, meaning that each makes its own decisions based on local observations and according to the models it has of the other agents. An agent $k$ is denoted by the tuple $\langle A_k, r_k \rangle$ comprising the agent's actions and goals. Each agent $k$ has a ToM (a model) of all other agents. We formally denote agent $k$'s model of another agent $g$ by $g^k = \langle A_g^k, r_g^k \rangle$. Each model thus contains the agent's beliefs about the actions and goals of each other agent. For the

purposes of our research, in this paper we assume agents have perfect models of other agents' goals and beliefs. However, they don't have a model of disruptions which prevents them from observing the future state of all agents completely.

Agents' decisions are based on a finite-horizon discounted reward scheme. Specifically, the goal of each agent $k$ is to select its actions at each time-step $t$ so as to maximize:

$$Q_k(s(t), \mathbf{a}(t)) = \mathbb{E}\left[\sum_{h=0}^{H} \gamma^t r_k(s(t+h))\right], \qquad (1)$$

where H is the planning horizon and $\gamma$ the discount factor. $Q_k(s(t), \mathbf{a}(t))$ represents the expected discounted value of executing some joint-action $\mathbf{a}$ in some state $s$ for agent $k$.

Agents decide by planning $H$ steps into the future. In our experiments this horizon is chosen according to the number of echelons and an assumed lead time, so that agents can see the effects of their decisions. During planning, agents reason about how they and all other agents choose their actions in each of the hypothetical future states $s(t+h)$, $h = 0, \ldots, H$. Specifically, at each planning time-step $t + h$, an agent models other agents as being greedy with respect to their reward function. As such, it selects an action $a_k^*(t+h)$ for each modeled agent $k$ that maximizes its expected discounted reward at time-step $t + h$ as given by:

$$a_k^*(t+h) = argmax_{a_k \in A_k} Q_k(s(t+h), \mathbf{a}^*(t+h)), \qquad (2)$$

where $\mathbf{a}^*(t+h) = \{a_k^*(t+h), a_1^k(t+h), \ldots, a_g^k(t+h), \ldots, a_k^k(t+h)\}$ is the expected optimal combination of all agents' actions. Each $a_g^k(t+h), g = 1, \ldots, K, g \neq k$ is the best response to $a_k^*$ that agent $k$ estimates agent $g$ will choose at hypothetical time-step $t + h$. Each best response $a_g^k(t+h)$ is calculated in a similar fashion by using agent $k$'s model of agent $g$, i.e., by choosing from the modeled action-set $A_g^k$ the action that maximizes $Q_g^k(s(t+h), \mathbf{a}(t+h))$ according to the modeled reward function $r_g^k$.

Agents therefore try to maximize their own reward while taking into account how other agents are also seeking to maximize their expected reward.

## Trustworthiness

At each time-step, a healthcenter updates its beliefs about the trustworthiness of each of its distributors based on the on-time delivery rate of that distributor. Formally, we update the *trustworthiness* that a healthcenter $h$ attributes to a distributor $d$ at each time-step $t$ according to:

$$T_{h,d,t} = (1 - \delta)T_{h,d,t-1} + \delta D_{h,d,t}, \qquad (3)$$

where $D_{h,d,t}$ is the calculated *on-time delivery rate* of distributor $d$ to healthcenter $h$ at time $t$, and $\delta$ is a *sensitivity factor* which determines how strongly the healthcenter reacts to the most recent changes in the distributor's on-time delivery rate. The higher this factor, the stronger the reaction of the healthcenter to recent observations is.

When each healthcenter orders from a distributor, the distributor promises a lead time. The on-time delivery rate of distributor $d$ to heathcenter $h$ at time $t$ is calculated using the last $l$ observations. Let us denote the actual amount of product that $h$ has received from $d$ at time $t$ by $R_{h,d,t}$ and the amount that $h$ expects to receive from $d$ at time $t$ by $E_{h,d,t}$. In calculating $E_{h,d,t}$, healthcenters account for order lead time, i.e., they expect to receive their order after the promised lead time. As such, we update $D_{h,d,t}$ at each observed time-step

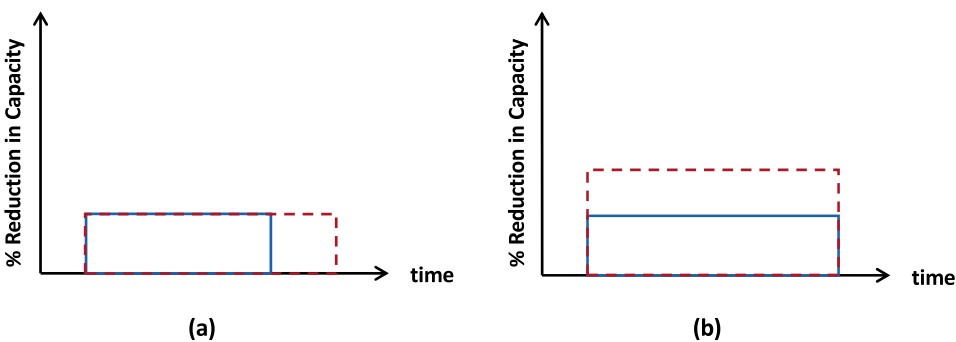

**Fig 2. Comparison between different disruption profiles.** Disruption A, dashed red line; disruption B, solid blue line; (a) disruption A has larger breadth compared to disruption B; (b) severity of disruption A is larger than severity of disruption B.

according to:

$$D_{h,d,t} = \sum_{k=t-l}^{t} R_{h,d,k}/E_{h,d,k} \qquad (4)$$

## Disruptions

Disruptions are unexpected events that interfere with the normal operation of some part of the supply chain for a certain duration of time. One of the most common disruptions in pharmaceutical supply chains is a reduction in the production capacity of a manufacturing facility [2]. These disruptions can vary in *severity*—the amount of reduction in production capacity—and *breadth*—the duration of time that the disruption lasts. We can see an illustration of different features of reduction in the production capacity disruption in Fig 2.

## Computational study setup

In the experiments that follow, the stylized model of the supply chain network is defined to be as small as possible in order to analyze and interpret the joint adaptation of agents' behaviors. Meanwhile, to study the consequences of an agent's decisions in the presence of other agents in the same echelon, more than one agent is needed in each echelon. Therefore, we modelled a 3-echelon supply chain with 2 agents in each echelon, as illustrated in Fig 3.

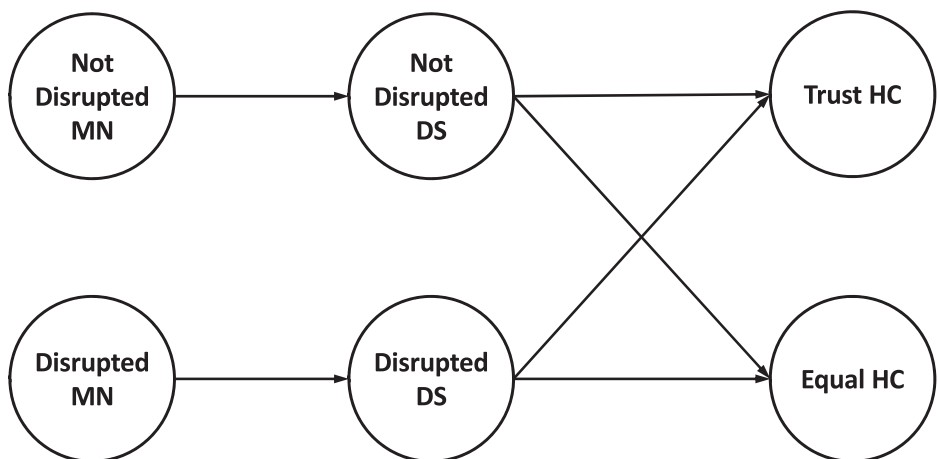

**Fig 3. The network structure of the supply chain.** The network consists of two manufacturers (*Not-Disrupted MN* and *Disrupted MN*), two distributors (*Not-Disrupted DS* and *Disrupted DS*) and two healthcenters (*Trust HC* and *Equal HC*).

**Table 2. Types of disruption profiles used in the experiments.**

| Disruption Type | Severity | Breadth |
|---|---:|---:|
| *Short* | 84% | 10 |
| *Moderate* | 42% | 20 |
| *Long* | 17% | 50 |

Agents are assumed to have a perfect model of other agents' states and actions, but they don't have a model of disruptions–importantly, this prevents agents from accurately modeling their future states and those of other agents during a disruption. At each time-step, healthcenters have constant patient demand. Given 3 echelons and an assumed lead time of 2 time-steps, we set the planning horizon $H = 6$ time-steps so that agents are able to reason about the effects of their decisions.

Each simulation starts at $t = 0$ and after a warm-up period the supply chain stabilizes, when the ordering and shipping levels become constant across the network. After stabilization, we simulate a disruption in at one of the manufacturers such that its production capacity decreases, as defined by a *severity* parameter, for a certain number of time-steps, which is defined by a *breadth* parameter. In the experiments that follow, we consider the three types of disruptions listed in Table 2. All disruptions result in the same total decrease in production capacity for the disrupted manufacturer, but have different temporal characteristics (i.e., the length of the disruption and the reduction in production capacity per time-step).

To examine the impact of the agents' changing beliefs on the performance of the supply chain, it is assumed that one of the healthcenters splits its order based on trustworthiness (*Trust HC*), while the other splits its order equally (*Equal HC*). Moreover, in order to isolate the effects of disruptions, coupled with the evolving agents' trustworthiness beliefs, each distributor is connected to only one manufacturer. This also ensures that they are not influenced by other interactions occurring in the network. One distributor (*Disrupted DS*) is supplied by the manufacturer that experiences a disruption (*Disrupted MN*). The other distributor (*Not-Disrupted DS*) is supplied by the manufacturer that is not disrupted (*Not-Disrupted MN*). Both healthcenters can place orders with both distributors.

We defined, analyzed, and compared five different scenarios whose corresponding parameters are provided in Table 3. In S1, we consider a scenario in which none of the agents perform ToM reasoning, i.e, they do not consider how other agents in the network react to their actions. In scenario S2, *Disrupted DS* performs ToM reasoning and changes the allocation decision calculation it uses, based on its expectations about the healthcenters' ordering behaviors. Allowing *Disrupted DS* to use ToM reasoning capability enables us to analyze the agent's response to other agents more realistically. In scenario S3, we study the destabilizing effect of the sensitivity factor $\delta$ by running simulations with settings similar to scenario S2 but with different values of $\delta$. Finally, in scenarios S4 and S5 we explore the role of exogenous factors in system outcomes, specifically disruption features, in combination with the sensitivity of agents.

**Table 3. The scenarios used in the experiments.** ToM refers to *Disrupted DS* performing Theory-of-Mind reasoning.

| Scenario | ToM | Disruption Type | Sensitivity $\delta$ |
|---|---|---|---:|
| S1 | No | *Short* | 0.5 |
| S2 | Yes | *Short* | 0.5 |
| S3 | Yes | *Short* | [0.025 − 0.5] |
| S4 | Yes | *Long* | [0.025 − 0.5] |
| S5 | Yes | *Moderate* | 0.5 |

## Results and discussion

### S1—No ToM reasoning

As listed in Table 3, in S1 *Trust HC* uses Eq 3 with sensitivity factor $\delta = 0.5$ to update the trustworthiness it attributes to the distributors and *Disrupted MN* experiences a *Short* disruption (Table 2). Moreover, none of the agents perform ToM reasoning.

We observe that when *Disrupted DS* does not have enough inventory to fulfill both healthcenters' demands, it allocates its inventory proportionally among them. *Trust HC* benefits from splitting its order based on trustworthiness and orders more from *Not-Disrupted DS*. Therefore, its overall cost is 14% less than *Equal HC*'s overall cost.

### S2—Effect of ToM reasoning by disrupted distributor

In this scenario the *Disrupted DS* performs ToM reasoning with a horizon of 6 time-steps. Surprisingly, *Disrupted DS*, accounting for both healthcenters' behavior, does not allocate its inventory proportionally, but instead prefers *Equal HC* more often than *Trust HC*. An understanding of *Disrupted DS*'s preference is provided by a closer examination of what happens during the disruption.

When the disruption occurs, the trustworthiness of *Disrupted DS* decreases and *Trust HC* orders more from *Not-Disrupted DS* (Fig 4(a)). *Not-Disrupted DS* reacts to this sudden change in demand by ordering more from its manufacturer. However, due to lead times, *Not-Disrupted DS* is not able to fulfill these larger orders on time for *Trust HC*. This results in a decrease in the trustworthiness of *Not-Disrupted DS* attributed by *Trust HC* (Fig 4(b)). Thus, even though the disruption affects only one of the distributors, the trustworthiness of both distributors decrease. Therefore, *Trust HC* repeatedly switches relative preferences for the distributors.

Consequently, agents that are not directly connected to the disrupted agents are destabilized as well and incur extra costs. Additionally, *Disrupted DS*, which performs ToM reasoning about healthcenters' behaviors, uses a preferential allocation since it can not profitably adjust to *Trust HC*'s oscillating ordering behavior. Thus, *Trust HC* is at a disadvantage compared to *Equal HC*.

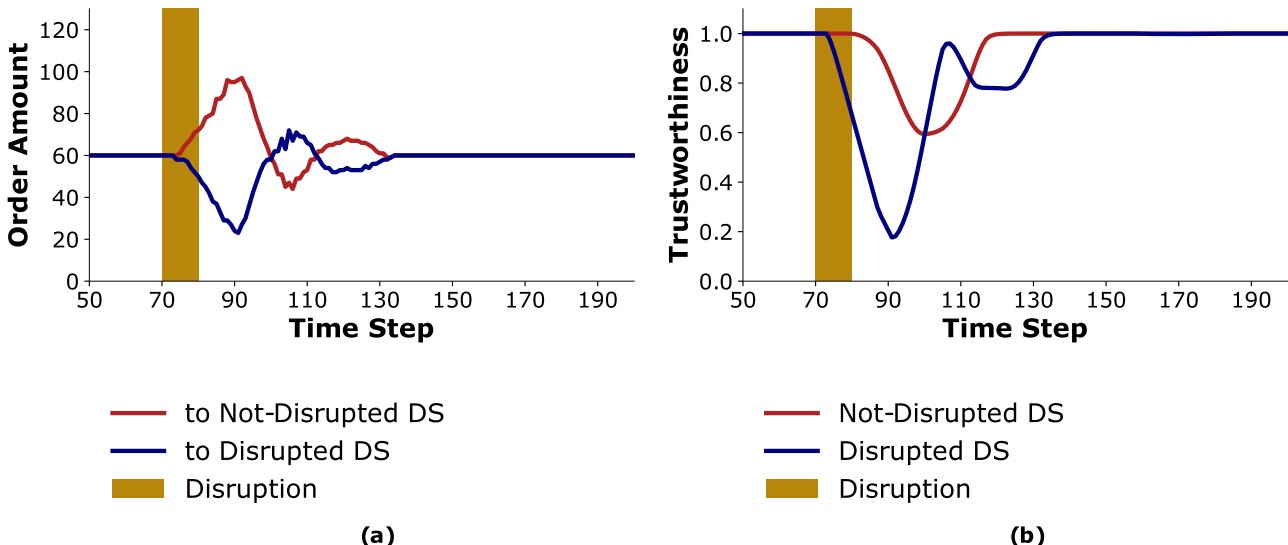

**(a)**　　　　　　　　**(b)**

**Fig 4. Results for S2.** (a) *Trust HC*'s order amounts to distributors; (b) *Trust HC*'s attributed trustworthiness to the distributors with $\delta = 0.5$.

**Table 4. Changes in agents' costs if one healthcenter uses trustworthiness to split the order compared to when both split orders equally when experiencing a *Short* disruption.**

| Agent | Change in Agent's Cost |
|---|---|
| *Trust HC* | 41% |
| *Equal HC* | -28% |
| *Not-Disrupted DS* | 7% |
| *Disrupted DS* | 12% |
| *Not-Disrupted MN* | 3% |
| *Disrupted MN* | 10% |
| *Overall Supply Chain* | 26% |

We compare this simulation with one where both healthcenters split their orders equally (neither healthcenter uses trustworthiness to split its order). As shown in Table 4, the overall supply chain cost is lower when both healthcenters split their orders equally. The effects are most significant for *Disrupted MN* and *Disrupted DS*. *Trust HC* is better off when it does not use trustworthiness to inform decision making, in which case *Disrupted DS* has no preference between healthcenters. However, *Equal HC* is worse off when both healthcenters split their orders equally, since *Equal HC* receives preferential treatment when *Trust HC* adjusts orders based on trustworthiness.

When we ran additional experiments with normally-distributed patient demands with different means and standard deviations, the results for demand with low variance were similar to Table 4. Details of these experiments are discussed in S1 Appendix.

## S3—Effect of sensitivity factor

As shown in the previous scenario, with $\delta = 0.5$, *Trust HC* is at a disadvantage. To study the effect of the sensitivity factor, we ran simulations equivalent to S2 with varying $\delta$ values. The results are depicted in Fig 5.

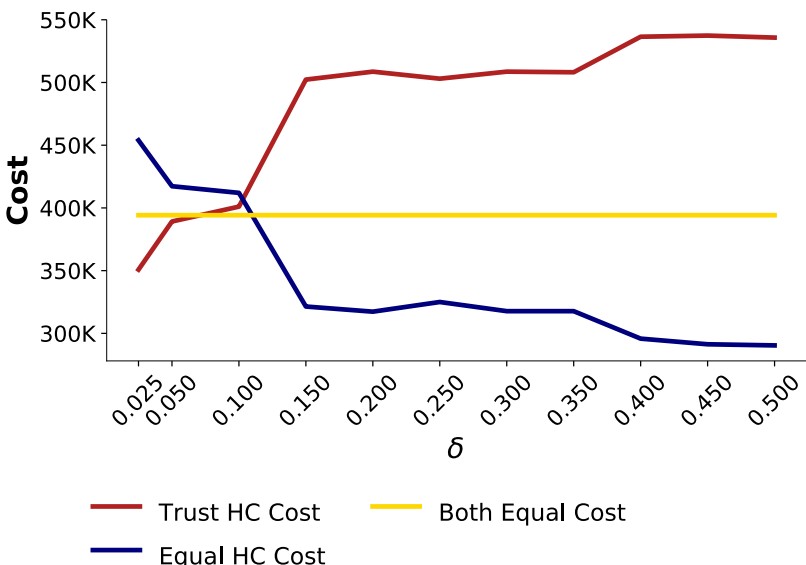

**Fig 5. Effect of sensitivity factor $\delta$ on healthcenters' costs when experiencing a *Short* disruption.**

For smaller $\delta$ values, *Trust HC* incurs fewer costs and therefore benefits from using trustworthiness. Fig 6(a) presents the trustworthiness attributed to *Disrupted DS* by *Trust HC* for $\delta$ = 0.4, 0.2, 0.05. Trustworthiness oscillates less for lower values of $\delta$. Further, Fig 6(b) shows that the variability in order amounts from distributors when $\delta$ = 0.05 is less than when $\delta$ = 0.5, as is shown in Fig 4(a). For small values of $\delta$, instead of oscillating between the distributors when the disruption occurs, *Trust HC* starts ordering more from *Not-Disrupted DS* and continues to do so until the aftermath of the disruption, at which point it starts ordering equally from both distributors.

We conclude that in order to optimize incorporating trust in its behavior, *Trust HC* should adjust its sensitivity factor, which will, in turn, also prevent the prolonged destabilization of the entire supply chain. As we will see in the next scenario, making such an intentional choice requires information about the disruption profile.

## S4—Effect of disruption profile

In this scenario, the *Disrupted MN* experiences a *Long* disruption rather than a *Short* one. Various sensitivity factors were simulated. The results from these simulations were compared with the results from the simulation in which both healthcenters split their orders equally.

The results for the *Long* disruption depicted in Fig 7 are very different from those with a *Short* disruption, depicted in Fig 5. Specifically, when the disruption is milder but longer, it is less costly for *Trust HC* if it updates trustworthiness with a $\delta$ as high as 0.5. In contrast, when facing a *Short* disruption, the same sensitivity factor of 0.5 destabilizes the whole supply chain and *Trust HC* incurs additional costs. Thus, the level of sensitivity $\delta$ that benefits the healthcenter under a certain disruption type can hurt the same healthcenter under a different type of disruption, as seen in the comparison of Tables 4 and 5. (See S1 Appendix for robustness of these results when patient demand is normally-distributed with different means and standard deviations.) As shown in Fig 7, *Trust HC* incurs a higher cost compared to *Equal HC*, but its cost is significantly lower than when it splits its order equally (Table 5).

This difference can be explained by examining changes in order amounts of *Not-Disrupted DS* during these disruptions, which is illustrated in Fig 8. An increase in orders to *Not-*

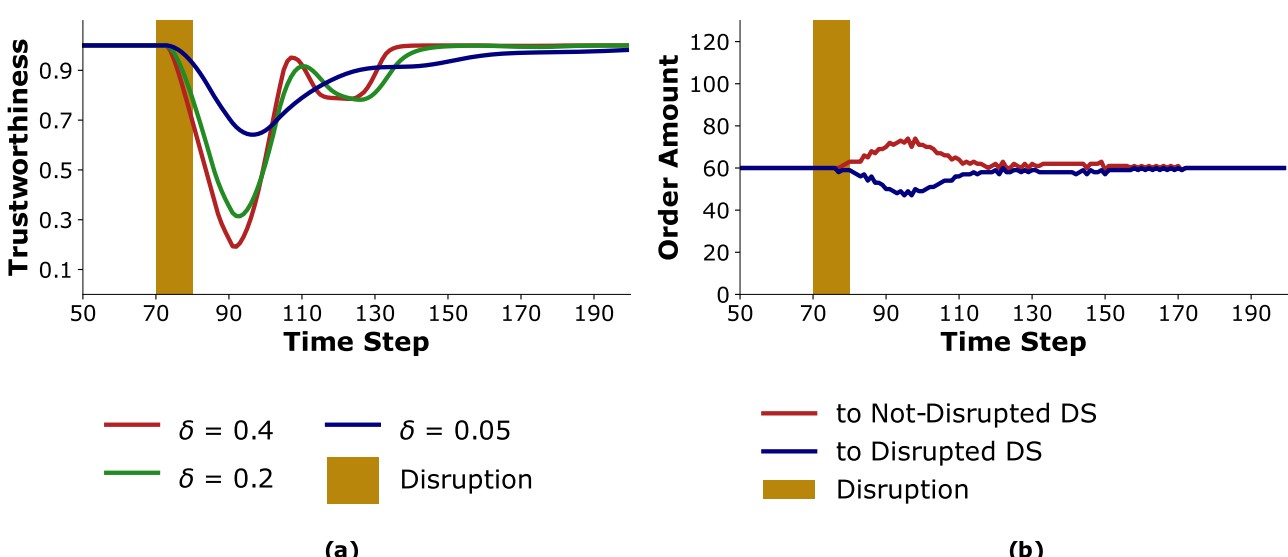

**Fig 6. (a).** *Trust HC*'s attributed trustworthiness to *Disrupted DS* with different sensitivity factors; (b) *Trust HC*'s order amount to both distributors when $\delta$ = 0.05.

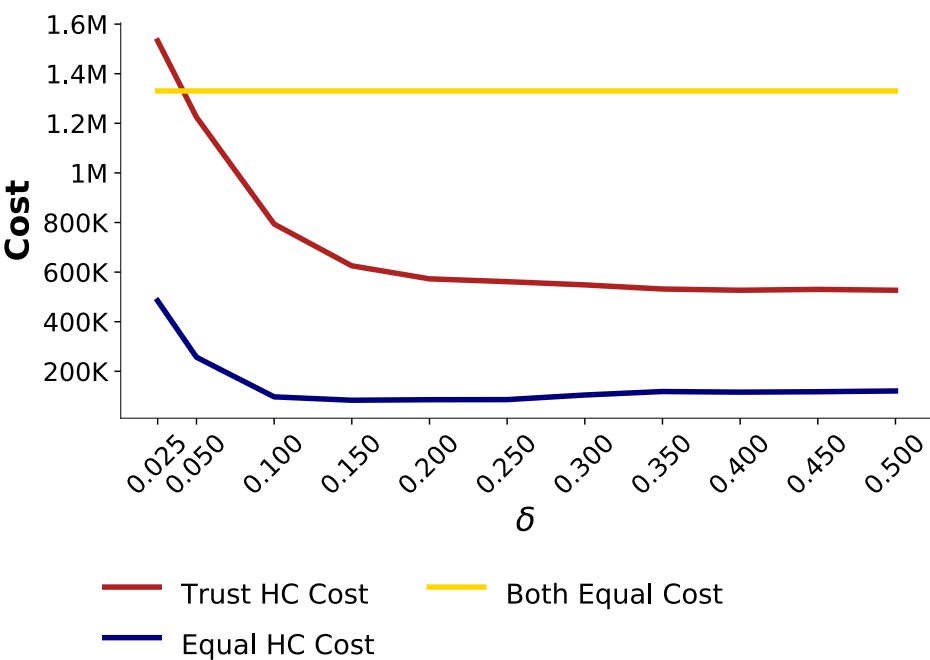

**Fig 7. Effect of sensitivity factor $\delta$ on healthcenters cost when experiencing a *Long* disruption.**

*Disrupted DS* requires increased production from *Not-Disrupted MN*, which in turn helps mitigate the effects of the disruption at *Disrupted MN* throughout the supply chain. However, due to the existence of lead times, it takes multiple time-steps for *Not-Disrupted DS* to update its order amount. For the *Short* disruption, by the time this change in production is achieved the disruption is already over and it does not help in mitigating the effects of the disruption. On the other hand, in a *Long* disruption, since the disruption is spread over a longer time period, increased orders by *Not-Disrupted DS* will lead to increased production by *Not-Disrupted MN* thereby bringing more product flow to the entire supply chain and mitigating the effects of the disruption.

## S5—Overall supply chain cost trajectory under different disruption profiles

In order to study how the overall supply chain cost trajectory changes with changes in the disruption profile, we also examine a scenario in which the disruption length is between the disruption lengths in the *Short* and *Long* disruptions, referred to as a *Moderate* disruption (see Table 2). As mentioned previously, the total decrease in production capacity of *Disrupted MN*

**Table 5. Changes in agents' costs if one healthcenter uses trustworthiness to order compared to when both split order equally when there is a *Long* disruption.**

| Agent | Change in Agent's Cost |
|---|---|
| *Trust HC* | -60% |
| *Equal HC* | -90% |
| *Not-Disrupted DS* | 14% |
| *Disrupted DS* | -139% |
| *Not-Disrupted MN* | 6% |
| *Disrupted MN* | -132% |
| *Overall Supply Chain* | -87% |

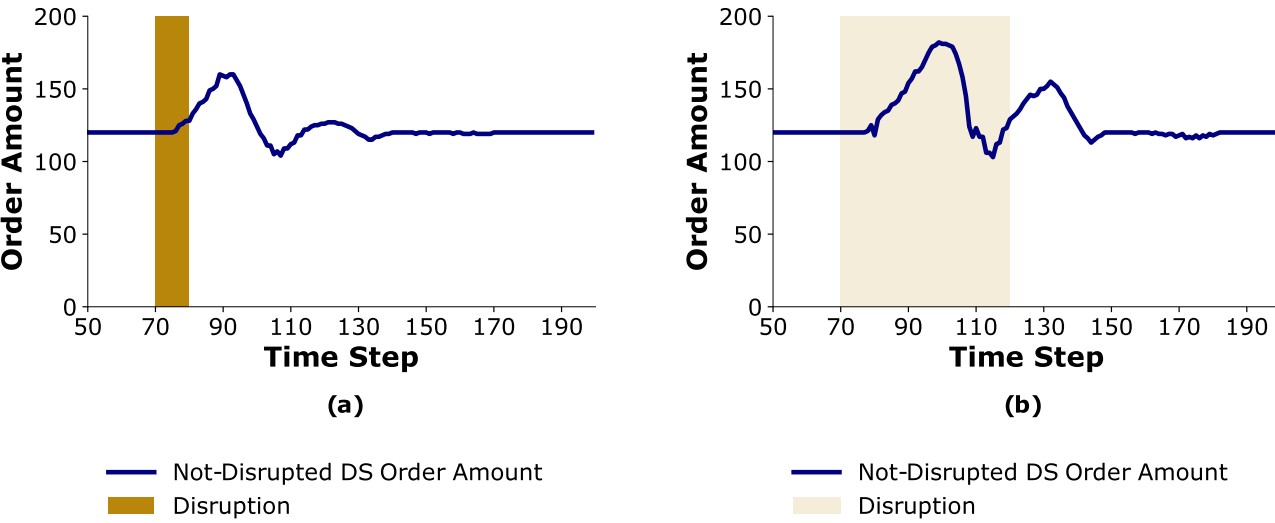

**Fig 8. Order amount of *Not-Disrupted DS*.** When (a) disruption is *Short*; b) disruption is *Long*.

is the same across all three disruption. In scenario S5, *Trust HC* uses $\delta = 0.5$ to update trust-worthiness. While it may be expected that the overall supply chain cost when experiencing a *Moderate* disruption would be less than the overall cost during a *Short* and greater than the overall cost observed during a *Long* disruption, this is not the case. As shown in Fig 9 the overall supply chain cost does not change linearly with the duration of the disruption. Instead, the overall supply chain cost in a system experiencing a *Moderate* disruption is greater than the overall costs from both *Short* and *Long* disruptions.

In order to explain this behavior we examine the changes in the trustworthiness that *Trust HC* attributes to the distributors under these three types of disruptions (see Figs 4(b), 10(b) and 10(d)). The trustworthiness that *Trust HC* attributes to *Disrupted DS* with *Moderate* and *Long* disruptions reduces to as low as zero. Additionally *Trust HC*'s order variability is higher with *Moderate* and *Long* disruptions compared to the *Short* disruption (compare Fig 10(a) and 10(c) with Fig 4(a)).

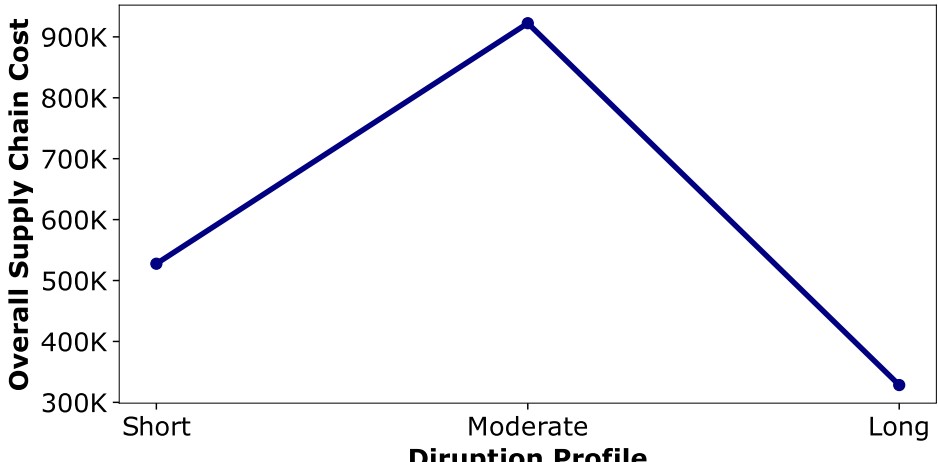

**Fig 9. Overall supply chain cost under different types of disruptions.**

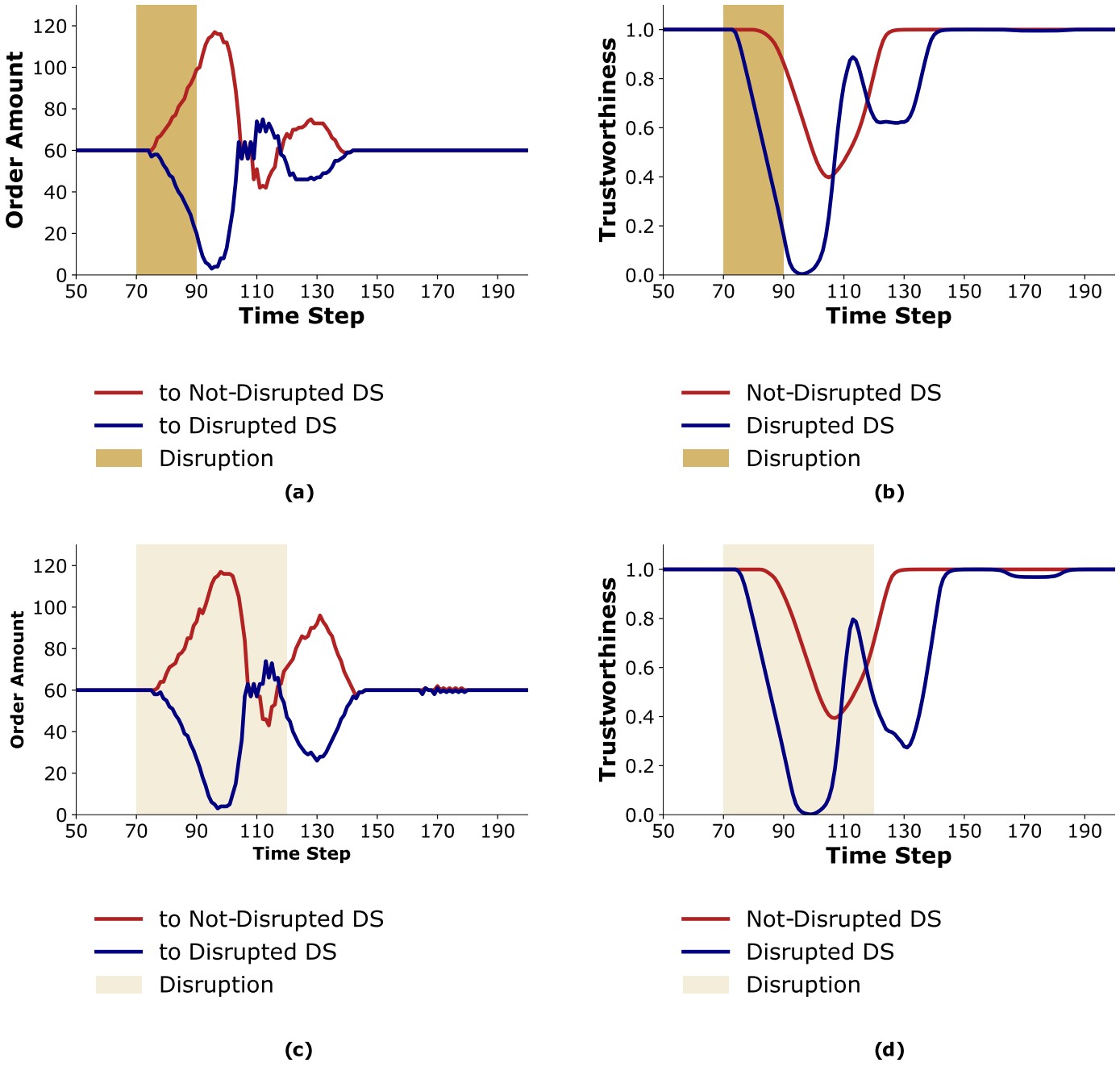

**Fig 10. Comparing order amounts and trustworthiness with *Moderate* and *Long* disruptions.** a) *Trust HC*'s order amounts to distributors with a *Moderate* disruption; b) *Trust HC*'s attributed trustworthiness to the distributors with $\delta$ = 0.5 and a *Moderate* disruption; c) *Trust HC*'s order amounts to distributors with a *Long* disruption; d) *Trust HC*'s attributed trustworthiness to the distributors with $\delta$ = 0.5 and a *Long* disruption.

As discussed in Scenario S4, in a system experiencing a *Long* disruption the additional amount of product flowing into the system (as a result of *Not-Disrupted DS* increasing its order amounts) happens during the time that disruption is still going on and it can help with mitigating the disruption. In contrast, with a *Moderate* disruption since the disruption is spread over a shorter time period, by the time the product flow in the system increases the disruption is already over, leading to higher inventory costs and having minimal effects on shortage costs (see Fig 11).

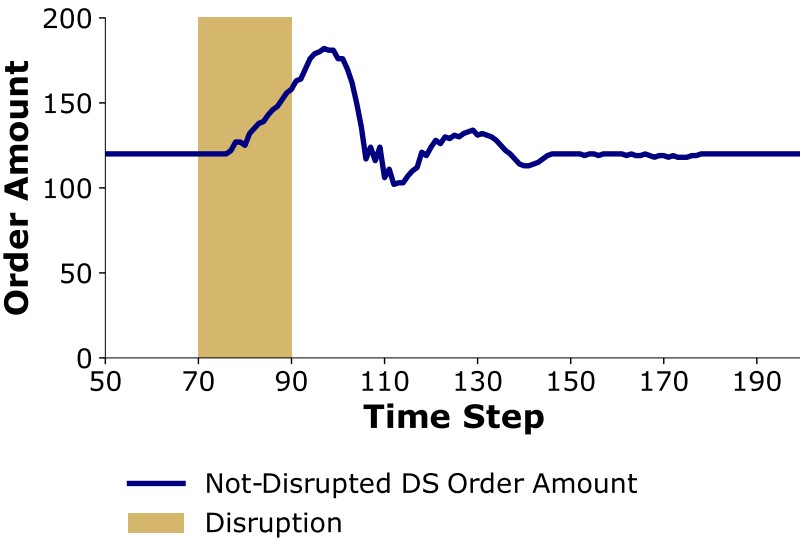

**Fig 11. Order amount of *Not-Disrupted DS* in face of *Moderate* disruption.**

To summarize, with *Moderate* disruption, the trustworthiness that *Trust HC* attributes to *Disrupted DS* goes down to zero which results in high order variability and at the same time the increase in product flowing to the system happens after the disruption is over. Therefore, the overall supply chain cost is higher compared to both *Short* and *Long* disruptions.

For the sake of completion, we included a table containing changes in all agents' costs if one healthcenter uses trustworthiness with $\delta = 0.5$ to order compared to when both split order equally when experiencing a *Moderate* disruption in S1 Table. The results from the robustness experiments for these results are included in S1 Appendix. In addition, we ran different simulations with different levels of sensitivity factors and report changes in the costs of both healthcenters with a *Moderate* in S1 Fig.

## Conclusion

Both industry leaders and healthcare providers agree that variability in orders and supply quantities are negatively affecting the performance of supply chains and the drug shortage crisis. In this paper, using a multiagent simulation, we aimed to identify the driving factors behind the cascading and prolonged destabilization of supply chains, by modeling agents with a ToM capability. The agents can plan ahead, accounting for other agents' expected reactions to their decisions in order to choose actions that maximizes their reward.

Having these more realistic models of human decision makers, our experiments showed that in order to obtain a thorough picture of the drivers of the oscillations, the interplay of the buyer and the suppliers' behaviors must be considered. In particular, a buyer's trust in a supplier changes its ordering behaviors, and the supplier in turn reacts to these changes by adapting its own behavior. Due to this co-adaptation, the entire supply chain performance degrades and the trust-sensitive buyer may end up being worse off.

However, this might be reversed under different settings where the buyer's trust-sensitive behavior may benefit itself and others. Specifically, in such settings, a trust-sensitive behavior induces non-disrupted distributors to order more from manufacturers. As a result, the amount of product flowing in the system increases, thereby helping in mitigating shortages. For this effect to transpire, the supplier needs to be transparent about the disruption, since different types of disruptions call for different optimal buyer behaviors.

Additionally, Our results argue that suppliers need to be able to estimate the duration and severity of the disruption, perhaps through better data collection and historical data analysis. Even if supply chain decision makers are not able to precisely estimate the disruption severity and duration, being aware of the above-mentioned drivers of variability in supply chains can aid in designing ordering policies, risk sharing mechanisms or even the conditions under which it is critical to share information.

Finally, we observed that changes in supply chain cost under different disruptions does not change linearly with the length of the disruption, when the total size of the disruption is held constant. This nonlinear relationship is driven by the complexity of the dynamics of the system, the interactions between suppliers and buyers, and the effects of lead times. These results support the need for future research examining the role of disruption features and temporal dynamics in supply chain shortages.

To assess the face validity of our simulations, the model, its assumptions, and results were presented in detail to four pharmaceutical informatics experts in two meetings using the methods outlined in [35]. The experts are members of the American Society of Health System Pharmacists. One of the experts is a long time consultant in the field, two are pharmacy PhDs, one with long time experience in pharmaceutical supply chain technology and the other an Executive Director of Health System Pharmacy Services at a large scale teaching hospital. The fourth expert is a leader in health information management.

The dynamics of agent behaviors, both in terms of order amounts and the choices of whom to order from and how to prioritize inventory allocation among healthcenters, and the dynamics of the trustworthiness measure were discussed and compared against the experts' experiences in the real world and well-studied supply chain simulations, such as the classic *Beer Game*. All experts agreed that: (i) the results of the simulations were representative of pharmaceutical supply chain decision makers' behaviors and (ii) analyzing how behaviors change based on trustworthiness changes brought new insights to the observed phenomena.

Decision makers in real-world supply chains face the challenge of maintaining mental models of other entities relying only on partial information. In future work, we will study supply chain networks where the agents have imperfect models of each other. We will investigate methods for potentially realizing *successful co-adaptation* based on enabling agents to anticipate their changing beliefs about each other and thereby explore ways to inform those beliefs.

## Supporting information

**S1 Appendix. Robustness of changes in cost if one healthcenter uses trustworthiness to order compared to when both split order equally under different disruption profiles.** (PDF)

**S1 Table. Changes in agents' costs if one healthcenter uses trustworthiness to order compared to when both split order equally under *Moderate* disruption.** (PDF)

**S1 Fig. Effect of sensitivity factor $\delta$ on healthcenters cost under *Moderate* disruption.** (EPS)

## Acknowledgments

We would like to thank Omid Mohaddesi and Casper Harteveld for their comments and feedback. We would also like to thank David Pynadath's help with PsychSim.

## Author Contributions

**Conceptualization:** Rozhin Doroudi, Pedro Sequeira, Stacy Marsella, Ozlem Ergun, Jacqueline Griffin.

**Data curation:** Rozhin Doroudi.

**Formal analysis:** Rozhin Doroudi.

**Funding acquisition:** Stacy Marsella, Ozlem Ergun, David Kaeli, Jacqueline Griffin.

**Investigation:** Rozhin Doroudi.

**Methodology:** Rozhin Doroudi, Pedro Sequeira.

**Project administration:** Jacqueline Griffin.

**Resources:** Stacy Marsella, Ozlem Ergun, David Kaeli, Jacqueline Griffin.

**Software:** Rozhin Doroudi, Pedro Sequeira, Rana Azghandi, Yifan Sun.

**Supervision:** Stacy Marsella, Ozlem Ergun, Jacqueline Griffin.

**Validation:** Rozhin Doroudi, Pedro Sequeira, Stacy Marsella, Ozlem Ergun, Rana Azghandi, David Kaeli, Yifan Sun, Jacqueline Griffin.

**Visualization:** Rozhin Doroudi.

**Writing – original draft:** Rozhin Doroudi, Pedro Sequeira.

**Writing – review & editing:** Rozhin Doroudi, Pedro Sequeira, Stacy Marsella, Ozlem Ergun, Jacqueline Griffin.

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
