## [Decision Letter · Decision Letter 0]

20 Dec 2019

PONE-D-19-27282

Effects of trust-based decision making in disrupted supply chains

PLOS ONE

Dear Ms. Doroudi,

Thank you for submitting your manuscript to PLOS ONE. After careful consideration, we feel that it has merit but does not fully meet PLOS ONE’s publication criteria as it currently stands. Therefore, we invite you to submit a revised version of the manuscript that addresses the points raised during the review process.

I recommend that it should be revised taking into account the changes requested by the reviewers. Since the requested changes include valuable comments, I would like to give you a chance to improve your manuscript.

We would appreciate receiving your revised manuscript by Jan 25 2020 11:59PM. To enhance the reproducibility of your results, we recommend that if applicable you deposit your laboratory protocols in protocols.io, where a protocol can be assigned its own identifier (DOI) such that it can be cited independently in the future. For instructions see: http://journals.plos.org/plosone/s/submission-guidelines#loc-laboratory-protocols

We look forward to receiving your revised manuscript.

Kind regards,

Baogui Xin, Ph.D.

Academic Editor

PLOS ONE

2.  We note that one or more of the authors are employed by a commercial company:  SRI International.

Reviewers' comments:

Reviewer's Responses to Questions

**Comments to the Author**

1. Is the manuscript technically sound, and do the data support the conclusions?

Reviewer #1: Yes

Reviewer #2: Yes

Reviewer #3: Yes

Reviewer #4: Yes

2. Has the statistical analysis been performed appropriately and rigorously? 

Reviewer #1: Yes

Reviewer #2: Yes

Reviewer #3: Yes

Reviewer #4: Yes

3. Have the authors made all data underlying the findings in their manuscript fully available?

Reviewer #1: Yes

Reviewer #2: Yes

Reviewer #3: Yes

Reviewer #4: Yes

4. Is the manuscript presented in an intelligible fashion and written in standard English?

Reviewer #1: Yes

Reviewer #2: Yes

Reviewer #3: Yes

Reviewer #4: Yes

5. Review Comments to the Author

Reviewer #1: To study a pharmaceutical supply chain with boundedly rational artificial decision makers capable of reasoning about the motivations and behaviors of others, authors use multi-agent simulations where each agent represents a key decision maker in a pharmaceutical supply chain. Their experiments reveal several counter-intuitive results showing how small, local disruptions can have cascading global consequences that persist over time. Authors also demonstrate how timing and scale of disruptions interacts with buyer’s sensitivity to trustworthiness.

It is good writing paper and is suitable for publications in PLOS ONE.

Reviewer #2: The manuscript is interesting and technically sound.

However, the English usage of the submitted paper need to be further polished, a careful reading of a native English speaker is necessary.

Reviewer #3: This paper is more interesting and now is presented in a good format. Still, it could be better to consider more scenarios for obtaining more comprehensive insights. If my concern is considered，this paper can be accepted for publication.

Reviewer #4: This paper reports on a multi-agent system for analyzing (combined) trust-based decisions and disruption impact on supply chain performance, cost for instance. The paper is well written and the explanation of the model is quite clear. The analysis of the example is also quite sound.

My only remark relates to the positioning of this research with regard to existing literature, I would suggest breaking down the introduction into two parts: 1) introduction and 2) state of the art and positioning. A quick search in two data bases pointed the following titles of published research works, which can benefit the improvement of the state of the art:

- A multi-agent systems approach for sustainable supplier selection and order allocation in a partnership supply chain

- An adaptive multi-agent system for cost collaborative management in supply chains

- Analysis of the performance of supply chains configurations using multi-agent systems

- Customer order fulfilment in mass customization context - An agent-based approach

6. PLOS authors have the option to publish the peer review history of their article (what does this mean?). If published, this will include your full peer review and any attached files.

Reviewer #1: No

Reviewer #2: No

Reviewer #3: No

Reviewer #4: No

---

## [Author Response · Author response to Decision Letter 0]

25 Jan 2020

We are extremely grateful to all reviewers for their comments and suggestions that helped improving the overall clarity of the paper. We have revised the entire manuscript in order to address all those comments and globally improve the readability and clarity of the presentation and discussions. Below we provide detailed replies to the reviewers’ comments and explain how the different suggestions were addressed and incorporated in the revised manuscript. 

Reviewer #1

We would like to thank the reviewer for the thoughtful comments. 

Reviewer #2

We would like to thank the reviewer for the thoughtful comments. The paper has been thoroughly proofread.

Reviewer #3

First, we would like to thank the reviewer for the thoughtful comments.

Regarding the comment about more scenarios to consider, we added a new scenario with a different disruption profile starting on page 12 of the manuscript. In this new scenario we examine how the overall supply chain cost trajectory changes with changes in disruption parameters. This resulted in uncovering new insights about nonlinear relationship between supply chain cost with the length of disruption, when the total size of the disruption is held constant. 

We also ran extra robustness simulations for scenarios 4 and 5. In these robustness simulations we consider stochastic patient demand and examine how using trustworthiness by one of the healthcenters versus not using trustworthiness by any of the healthcenters affects supply chain agents’ cost.

Reviewer #4

We would like to thank the reviewer for the thoughtful comments.

Following the suggestion to add a separate section for the literature review, we added a “State of the art” section starting on page 3 of the manuscript. In this section we positioned our research within the multiagent system literature. In particular, we highlighted papers that, similarly to our own, consider supplier-buyer interaction in a supply chain. The corresponding additional references are as follows:

• Cachon GP, Lariviere MA. Capacity choice and allocation: Strategic behavior and supply chain performance. Management science. 1999 Aug; 45(8):1091-108.

• Durango-Cohen EJ, Li CH. Modeling supplier capacity allocation decisions. International Journal of Production Economics. 2017 Feb 1; 184:256-72.

• Chen CM, Thomas DJ. Inventory Allocation in the Presence of Service-Level Agreements. Production and Operations Management. 2018 Mar; 27(3):553-77.

• Kawtummachai R, Van Hop N. Order allocation in a multiple-supplier environment. International Journal of Production Economics. 2005 Jan 8; 93:231-8.

• Wooldridge M, Jennings NR. Intelligent agents: Theory and practice. The knowledge engineering review. 1995 Jun; 10(2):115-52.

• Swaminathan JM, Smith SF, Sadeh NM. Modeling supply chain dynamics: A multiagent approach. Decision sciences. 1998 Jul; 29(3):607-32.

• Moyaux T, Chaib-Draa B, D'Amours S. Supply chain management and multiagent systems: an overview. Multiagent based supply chain management 2006 (pp. 1-27). Springer, Berlin, Heidelberg.

• Lee JH, Kim CO. Multi-agent systems applications in manufacturing systems and supply chain management: a review paper. International Journal of Production Research. 2008 Jan 1; 46(1):233-65.

• Kimbrough SO, Wu DJ, Zhong F. Computers play the beer game: can artificial agents manage supply chains? Decision support systems. 2002 Jul 1; 33(3):323-33.

• Ghadimi P, Toosi FG, Heavey C. A multi-agent systems approach for sustainable supplier selection and order allocation in a partnership supply chain. European Journal of Operational Research. 2018 Aug 16; 269(1):286-301.

• Fu J, Fu Y. An adaptive multi-agent system for cost collaborative management in supply chains. Engineering applications of artificial intelligence. 2015 Sep 1; 44:91-100.

• Giannakis M, Louis M. A multi-agent based framework for supply chain risk management. Journal of Purchasing and Supply Management. 2011 Mar 1; 17(1):23-31.

We would also like to thank the reviewer for the suggested papers. We carefully read them and decided to add in the “State of the art” section, on page 4 of the manuscript, the ones that were most closely related to our research, namely:

• Ghadimi P, Toosi FG, Heavey C. A multi-agent systems approach for sustainable supplier selection and order allocation in a partnership supply chain. European Journal of Operational Research. 2018 Aug 16; 269(1):286-301.

• Fu J, Fu Y. An adaptive multi-agent system for cost collaborative management in supply chains. Engineering applications of artificial intelligence. 2015 Sep 1;4 4:91-100.

---

## [Editor Report · Decision Letter 1]

30 Jan 2020

Effects of trust-based decision making in disrupted supply chains

PONE-D-19-27282R1

Dear Dr. Doroudi,

We are pleased to inform you that your manuscript has been judged scientifically suitable for publication and will be formally accepted for publication once it complies with all outstanding technical requirements.

With kind regards,

Baogui Xin, Ph.D.

Academic Editor

PLOS ONE
---

## [Editor Report · Acceptance letter]

7 Feb 2020

PONE-D-19-27282R1 

Effects of trust-based decision making in disrupted supply chains 

Dear Dr. Doroudi:

I am pleased to inform you that your manuscript has been deemed suitable for publication in PLOS ONE. Congratulations! Your manuscript is now with our production department. 

With kind regards,

on behalf of

Prof. Baogui Xin 

Academic Editor

PLOS ONE